# Contribution of the Nrf2 Pathway on Oxidative Damage and Mitochondrial Failure in Parkinson and Alzheimer’s Disease

**DOI:** 10.3390/antiox10071069

**Published:** 2021-07-02

**Authors:** Francisca Villavicencio Tejo, Rodrigo A Quintanilla

**Affiliations:** Laboratory of Neurodegenerative Diseases, Instituto de Ciencias Biomédicas, Facultad de Ciencias de la Salud, Universidad Autónoma de Chile, Santiago 8910060, Chile; franvilla.tejo@gmail.com

**Keywords:** Alzheimer’s disease, Parkinson’s disease, mitochondria, neurodegeneration, Nrf2 neuroprotection

## Abstract

The increase in human life expectancy has become a challenge to reduce the deleterious consequences of aging. Nowadays, an increasing number of the population suffer from age-associated neurodegenerative diseases including Parkinson’s disease (PD) and Alzheimer’s disease (AD). These disorders present different signs of neurodegeneration such as mitochondrial dysfunction, inflammation, and oxidative stress. Accumulative evidence suggests that the transcriptional factor nuclear factor (erythroid-derived 2)-like 2 (Nrf2) plays a vital defensive role orchestrating the antioxidant response in the brain. Nrf2 activation promotes the expression of several antioxidant enzymes that exert cytoprotective effects against oxidative damage and mitochondrial impairment. In this context, several studies have proposed a role of Nrf2 in the pathogenesis of PD and AD. Thus, we consider it important to summarize the ongoing literature related to the effects of the Nrf2 pathway in the context of these diseases. Therefore, in this review, we discuss the mechanisms involved in Nrf2 activity and its connection with mitochondria, energy supply, and antioxidant response in the brain. Furthermore, we will lead our discussion to identify the participation of the Nrf2 pathway in mitochondrial impairment and neurodegeneration present in PD and AD. Finally, we will discuss the therapeutic effects that the Nrf2 pathway activation could have on the cognitive impairment, neurodegeneration, and mitochondrial failure present in PD and AD.

## 1. Introduction

In recent years, the interest to study the contribution of the Nrf2 pathway on PD and AD has become a topic of great interest [1,2]. Several studies have suggested that Nrf2 activation could be proposed as a novel strategy to reduce oxidative damage and mitochondrial dysfunction present in PD and AD [3,4]. Oxidative and mitochondrial damage occurs in the early stages of these disorders [5], suggesting that these defects could play a role in the progression of these neurodegenerative diseases (NDs) [6,7].

Mitochondria are essential organelles involved in energy supply and represent a significant source of reactive oxygen species (ROS) production in the cells [8]. The central nervous system (CNS), particularly the brain, presents high energy requirements, making it more vulnerable to mitochondrial dysfunction and oxidative stress [9]. Decreased ATP production is a common hallmark in PD and AD and can also be caused by defects in mitochondrial function such as respiratory imbalance and mitochondrial uncoupling [10].

Nrf2 activation increases the expression of antioxidant enzymes such as NAD(P)H dehydrogenase quinone 1 (NQO1), heme oxygenase 1 (HO-1), and others [9]. Consequently, these actions improve mitochondrial function and ATP synthesis and prevent oxidative damage [11]. Additionally, current studies suggests an important role of Nrf2 regulating mitochondrial biogenesis, dynamics, and mitophagy [11]. Interestingly, Nrf2 activity declines during aging [12], causing a progressive loss of glutathione (GSH) synthesis in aged-rats of 24–28 months of age [13]. The loss of GSH regulation and the existence of a pro-oxidant state in aging cells indicates that the cellular antioxidant defenses could be progressively affected by senescence [14]. In this context, it has been suggested that the Nrf2/ARE pathway is affected by aging and NDs [15]. For example, a meta-analysis of AD and PD gene expression of different tissues revealed that from 54 affected genes, 31 were downregulated genes containing ARE (antioxidant response elements) [16]. Studies from Branca and colleagues also performed in APP/PS1 mice, a transgenic mice strain that produces an excess of β-amyloid plaques and cognitive impairment, showed a decrease in Nrf2 activity and significant reduction in HO-1 levels concomitantly with AD pathology and cognitive impairment [17]. Additionally, studies related to PD pathology showed that Nrf2 expression was significantly impaired in nigral dopaminergic neurons of PD patients [18]. More importantly, this reduction of dopaminergic neurons and inflammatory-mediated microglia activation were enhanced in Nrf2 (−/−) knock out mice [19].

Finally, a growing body of evidence suggests that the activation of Nrf2 pathway can be considered as a valid therapeutic target against PD and AD [20]. Indeed, several drugs or natural compounds with Nrf2 activity have been used for proof-of-concept studies indicating that activation of this pathway could provide a positive outcome against PD [21] and AD [4].

In this review, we present an overview of the Nrf2 pathway and its contribution to oxidative damage and mitochondria failure present in AD and PD. We also discuss evidence that suggests Nrf2 activation as a promising target to ameliorate the neurodegenerative changes present in these disorders.

## 2. Nrf2 Pathway

### 2.1. Regulation of Nrf2 Pathway by Protein Stability

Nrf2 is a transcriptional factor encoded by the gene *NFE2L2* related to the Cap’n’collar family of transcription factors that regulates the basal and stress-inducible expression of over 250 genes that contain ARE sequence (core sequence: (A/G)*TGA(G/C)TCA*GCA) [22], in their promoters [23,24,25]. These genes constitute a defensive response against oxidative agents including genes that encode for HO-1, NQO1, glutathione S-transferase (GST), glutamate-cysteine ligase (GCL), glutathione peroxidase (GPx), thioredoxin reductase (TXNRD)1, thioredoxin (TXN)1, and reduced glutathione (GSH) [26,27,28,29]. The activation of the Nrf2-ARE pathway results in increasing cellular energy and redox potential, which reduce oxidative damage [30].

The regulation of Nrf2 occurs mainly by two specific mechanisms: the Kelch-like ECH-associated protein 1 (Keap1), and the participation of b-TrCP with the glycogen synthase kinase-3β (GSK)-3β [31] (Figure 1). In normal cells, Keap1 and the E3 ubiquitin ligase substrate adaptor regulate the Nrf2 protein levels in a redox-dependent manner [32,33]. The protein Keap1 forms a homodimer responsible for sequestering Nrf2 in the cytosol and rendering it inactive [34]. Keap1 also binds Cullin 3 (Cul3), which forms a core with the E3 ubiquitin ligase complex through an association with Ring-box1 protein (Rbx1, also called Roc1) [32,33,35,36]. In homeostatic conditions the Keap1–Cul3-Rbx1 complex can ubiquitinate Nrf2 and target it for proteasome degradation [37,38]. In response to electrophiles and oxidants, that identify and chemically modify specific cysteine residues of Keap1, Nrf2 ubiquitination is inhibited [39] (Figure 1).

Nrf2 is also subjected to degradation mediated by GSK-3β and TrCP-dependent Cul1-based ubiquitin ligase [40]. GSK-3β phosphorylates Nrf2 at Ser334–338 protein residues, which creates a degradation domain that is recognized by β-TrCP and tagged for proteasomal degradation by the Cullin 1 (Cul1) and Rbx1 complex [40,41]. These actions promote Nrf2 nuclear exclusion and degradation by a Keap1-independent manner [42]. (Figure 1). Under oxidative stress conditions such AD or PD, Nrf2 modulates its activity through Fyn protein phosphorylation induced by GSK-3β [43,44] Phosphorylated Fyn protein translocates and accumulates in the nucleus and phosphorylates Nrf2, leading to its nuclear export, ubiquitination, and degradation [45,46] (Figure 2).

In a typical redox environment, minimal Nrf2 activity maintains the basal expression of some ARE-driven genes [47]. In these conditions, the thiol groups of the redox-sensitive Cys residues located in Keap1 are not modified by the action of electrophiles and pro-oxidant environment present in the cells [48]. Thus, under these conditions, the degradation of Nrf2 occurs mainly via Keap1–Cul3/Rbx1 complexes [43,45,49].

### 2.2. Transcriptional and Post-Transcriptional Regulation of Nrf2

In addition to the regulation of the Nrf2 protein stability, the genetic products from the Nrf2 pathway can be controlled by post-transcriptional modifications and the availability of binding partners [44]. In this case, the transcriptional factors involved include aryl hydrocarbon receptor (AhR), nuclear factor kappa-light-chain-enhancer of activated B cells (NF-κB), and Nrf2 itself [50]. AhR induces Nrf2 activity in response to polycyclic aromatic hydrocarbon exposure [51]. AhR and Nrf2 signaling modulate the expression of genes that affect the metabolism of xenobiotics [51]. Nrf2 gene transcription can be modulated by AhR activation and this signaling is also present in the opposite direction, suggesting that the AhR gene is directly affected by Nrf2 [52]. The response element phase II genes and xenobiotic response element (XRE) may recognize these transcription factors, which are present in the regulatory domains of the same target genes such NQO1 [53]. Furthermore, the Nrf2 gene promoter contains a binding site for NF-κB subunits p50 and p65, which are involved in the transactivation of the Nrf2 gene [54]. NF-κB can trigger Nrf2 signaling and participate in the decrease of NF-κB, indicating an interesting crosstalk between these two pathways [55]. In addition, p65, a canonical NF-κB subunit, can exert a negative regulation on ARE-linked gene expression [56]. p65 participates in increasing the abundance of Keap1 levels, thus decreasing Nrf2 activity [56]. Additionally, the treatment with LPS in a Nrf2 (−/−) mice model with amyotrophic lateral sclerosis (ALS) showed elevated NF-κB activity and an increase in cytokine production that contributed to astrogliosis, neuronal death, and demyelination of neuronal axons [57]. Nrf2 and NF-κB are tightly regulated by redox factors and the lack of Nrf2 is related to an increased oxidative stress, conductive to a high cytokine production and leading to the NF-κB activation in an oxidative environment [58].

Under stress conditions or under the effect of natural compounds, Nrf2 stabilizes and translocates to the nucleus, where it binds (as a heterodimer member of the small Maf family of transcription factors) to the ARE/EpRE sequences in the promoter of its target genes [59] (Figure 2). The Nrf2 cytoprotective response has been described in various mammalian tissues, cultured cells, and other model organisms such as Drosophila melanogaster and Caenorhabditis Elegans [60,61]. Interestingly, these organisms have shown similar antioxidant systems to mammals, indicating that the Nrf2 pathway represents an evolutionarily conserved defense mechanism [62].

Growing evidence has shown that Nrf2 is involved in the regulation of the unfolded protein response (UPR), which is triggered by the accumulation of misfolded proteins in the endoplasmic reticulum [63,64]. Alternatively, studies showed that during endoplasmic reticulum stress, Nrf2 is also regulated by ubiquitination [65]. Nrf2 is ubiquitinated and degraded in a process mediated by the E3 ubiquitin ligase, Hrd1, p62/sequestosome-1 (p62/SQSTM1), and p21Cip1/WAF1, which interfere with the formation of the Keap1–Nrf2 ubiquitination complex regulating the abundance of Nrf2 [66,67]. Thus, the use of electrophilic compounds that target the Nrf2/Keap1 axis could increase the transcriptional activity of Nrf2 and strengthen cellular antioxidant defenses.

Importantly, Nrf2 contributes to removing misfolded or damaged proteins by regulating proteasome degradation [68,69]. In this regard, Nrf2 upregulates the expression of several proteasome subunits, protecting the cell from the accumulation of toxic proteins [70]. Twenty proteasome and ubiquitination-related genes are regulated by Nrf2 through the Keap1-Nrf2-ARE signaling pathway in agreement with a wide microarray analysis from liver RNA that was set up with Nrf2 natural activators, sulforaphane, or inducer 3*H*1,2-dithiole-3-thione (D3T) [71,72]. These studies showed an enhanced activity of the 26S proteosome promoted by Nrf2 activators [69].

### 2.3. Nrf2 and the Antioxidant Response in the Brain

Under oxidative stress, brain cells adjust their metabolism and gene expression to maintain redox homeostasis by activating Nrf2 and other stress response pathways [30,73,74]. For example, Kovac and colleagues found that the Keap1–Nrf2 pathway modulates ROS production via NADPH oxidase in primary neurons and brain explant slice cultures [75]. ROS production in Nrf2 (−/−) cells was significantly increased compared to WT cells, indicating that Nrf2 participates in regulating the redox and the intermediary metabolism [75,76]. Furthermore, Nrf2 can induce the expression of the principal enzymes implicated in NADPH generation: isocitrate dehydrogenase 1 (IDH-1), glucose-6-phosphate dehydrogenase (G6PD), malic enzyme 1 (ME-1), and 6-phosphogluconate dehydrogenase (PGD) [76,77,78]. In the brain, Nrf2 also coordinates the activity of thioredoxins (TRXs) and peroxiredoxins (PRXs) necessary for the removal of ROS in the mitochondria [79]. Nrf2 increases the inducible expression of GSH biosynthesis enzymes and its regeneration enzyme GSH reductase, which plays a crucial role in maintaining the mitochondrial GSH pool [80,81]. GSH peroxidases (GPx1 and GPx4) detoxify superoxide-derived hydrogen peroxide to water using GSH and NADPH [82]. Additionally, Nrf2 activation is associated with the induction of mitochondrial antioxidant enzymes like thioredoxin reductase-2 (Txnrd2), peroxiredoxin 3 (Prdx3), 5 (Prdx5), GPx1, and mitochondrial superoxide dismutase 2 (SOD2) [83] However, in this process, the regulatory mechanisms induced by Nrf2 are not entirely elucidated [83,84,85,86].

In this context, Lee and colleagues have shown that increased Nrf2 activity by specific activators enhanced resistance to mitochondrial toxins such as the complex I inhibitor, rotenone, or the complex II inhibitor, 3-nitropropionic acid (3-NP) in primary cortical astrocytes from Nrf2 (−/−) mice [87]. Complementary, these studies showed that astrocytes from Nrf2 (−/−) treated with 3-NP showed a hypersensibility to this drug inducing neurodegeneration and astrocyte activation [88,89]. Likewise, the genetic ablation of Nrf2 reduces the constitutive and inducible expression of cytoprotective genes, thereby enhancing the sensitivity of neurons and astrocytes to the oxidative damage [77].

### 2.4. Nrf2 and Mitochondrial Function in the Brain

Mitochondria function is essential for cellular bioenergetics, biosynthesis, and other related process [88]. During respiration and oxidative phosphorylation, mitochondria use oxygen to generate ATP by the electron transport chain (ETC) [89,90]. In this process, there is an inevitable generation of ROS [91], which also represents a key signaling molecules [92]. ROS excess is harmful by modifying macromolecules such as proteins, lipids, and DNA, thereby disrupting cellular homeostasis and inducing mitochondrial impairment [89]. Mitochondrial sensitivity to oxidative stress is strongly implicated in the pathophysiology of many diseases and disorders including those affecting the CNS [92,93,94,95]. Vulnerable mitochondrial sites for oxidative stress include metabolic enzymes, electron transport chain, oxidative phosphorylation, DNA, RNA, membrane lipids, Ca^2+^ handling proteins, and mitochondrial permeability induced by the opening of the mitochondrial permeability transition pore (mPTP) [96]. Opening of mPTP causes mitochondrial depolarization, uncoupling, and mitochondrial metabolite release including pyridine nucleotides and glutathione [97].

An interesting connection between Nrf2 and mitochondria was reported by Lo and colleagues [98]. Their studies showed that Keap1 associates with phosphoglycerate mutase 5 (PGAM5), a protein phosphatase related to mitochondria homeostasis, mitophagy, and cell death [98,99]. Additionally, recent reports indicate that the expression of PGMA5 results in decreased mitochondrial movement, which is particularly essential for the transport of these organelles along the axon [100]. Furthermore, several studies have suggested a link between the Nrf2 pathway and mitochondrial function, which we will discuss in the next sections.

#### 2.4.1. Mitochondrial Bioenergetics

Growing evidence indicates that Nrf2 influences primary metabolism and bioenergetics [101,102]. For instance, the genetic suppression of Nrf2 reduced the expression levels of the malic enzyme, glucose-6-phosphate dehydrogenase (G6PD), transaldolase (TAL), and transketolase (TKT) [28,76]. These enzymes participate in the biosynthesis of primary metabolites including NADPH, glutathione, ribose-5-phosphate (fatty acids), and erythrose-4-phosphate (aromatic amino acids) [76]. Additionally, other studies have shown that genetic ablation of Nrf2 leads to mitochondrial failure, defects in fatty acid oxidation, respiration, and ATP production [103,104]. Moreover, Holmström et al. studied the role of Nrf2 and its repressor Keap-1 on mitochondrial bioenergetics using isolated mitochondria derived from primary neuronal culture of Nrf2 (−/−) mice [105]. They showed defective respiration, mitochondrial uncoupling, and decreased ATP levels in Nrf2 (−/−) neuronal cultures. Interestingly, when the Nrf2/ARE pathway is genetically activated, it restores mitochondrial membrane potential and the respiration rate levels, confirming that the Nrf2-Keap1 pathway modulates cellular energy metabolism through mitochondrial substrate availability.

#### 2.4.2. Mitochondrial Biogenesis

One of the first links between Nrf2 and mitochondrial biogenesis was provided by Piantadosi et al. [106]. They showed that Nrf2 activation resulted in transcriptional upregulation of Nrf1/alpha-PAL and the transactivation of mitochondrial biogenesis genes that encode for NADH dehydrogenase subunit 1 (ND1) and cytochrome c oxidase (COX) subunit I [106]. In addition, other studies have suggested that Nrf2 also influences mitochondrial biogenesis by activating the mitophagy process under stress conditions [75]. This process maintains the organelle integrity by selectively removing damaged mitochondria [107]. One of the critical components for this process is the autophagy adaptor protein sequestosome-1 (SQSTM1/p62) [108]. p62 acts as an adaptor that binds protein aggregates that are conducted to ubiquitination and delivers them to autophagosomes [109]. Interestingly, p62 contains an interacting region (KIR) domain of Keap-1 (349-DPSTGE-354) [110,111], resembling the Keap1-interacting ETGE motif in the Neh2 domain of Nrf2, allowing p62 to sequester Keap1, preventing the ubiquitylation of Nrf2, and favoring its activation [111,112]. Furthermore, Nrf2 promotes mitochondrial biogenesis through the activation of the family of peroxisome proliferator-activated receptor coactivators (PGC) including PGC-1α and PGC-1β, which are involved in the process of mitochondrial biogenesis [113]. Complementary studies showed that Nrf2 could interact with PGC1α, inducing the expression of mitochondrial SOD2, preserving mitochondrial mass, and reducing oxidative damage [1,104].

### 2.5. Nrf2 and Neuroinflammation

Another important connection between Nrf2 and NDs is related to the neuroinflammation in pathologies like AD and PD [114,115]. Neuroinflammation is a process characterized by changes in the morphology of glial cells including both astrocytes and microglia as well as the inflammatory cytokine release (IL-1β, IL-6, and TNF-α) [116,117]. The prolonged and chronic inflammatory responses in the CNS lead to the augmented release of inflammatory mediators and oxidative stress, thereby maintaining neuroinflammation cascades and accelerating neuronal dysfunction [118].

In this context, ARE-regulated Nrf2-dependent genes are activated in stressed astrocytes, suggesting an interesting link between the impairment of these cells and the role of oxidative stress in neurodegeneration [119]. Additionally, Nrf2 activation maintains redox homeostasis in microglia [114]. For example, the release of CX3CL activates the Nrf2 signaling in microglial cells [116,120]. CX3CR is the receptor for the chemokine fractalkine (CX3CL1), which is a critical pathway for microglia-neuron crosstalk [121]. Interestingly, other studies have investigated the crosstalk between CX3CR1/Nrf2 in the context of tauopathies and neuroinflammation [116]. Here, they observed a significant decrease in the mRNA levels of Nrf2 and its related genes in primary microglia cultures from Cx3cr1 (−/−) mice [116]. Indeed, Nrf2 has been considered as a potential candidate for pharmacological targeting to ameliorate neurodegenerative changes induced by neuroinflammation [122].

## 3. Neurodegeneration in PD

PD ranks second among the diseases with the highest prevalence occurrence worldwide [123]. While PD has been widely investigated, the primary causes remain unsolved, even though several risk factors have been identified including age, environmental toxin exposure, and genetic mutations [124,125]. One of the most important pathological hallmarks present in PD is the selective loss of dopaminergic neurons in the substantia nigra (SN) pars compacta and the loss of dopamine nerve terminals projecting to the striatum [126]. Another hallmark in PD is the formation of protein inclusions within the cytosol known as Lewy bodies, mainly constituted by aggregates of the protein α-synuclein (α-Syn) [127,128]. Additionally, clinical/pathological studies showed that the progressive neuronal death from the ventrolateral region of the SN represents a clear sign of the PD presence in the brain [129].

Several studies have indicated that the genetic and molecular causes involved in PD are intrinsically related to the mitochondrial impairment, oxidative stress, and endo-lysosomal system dysfunction [130,131]. Additionally, the accumulation of aberrant or misfolded proteins and the impairment of the ubiquitin-proteasome system contributes to the pathogenesis of sporadic and familial PD [130,131]. In addition, some studies have shown a trilateral correlation between mitochondrial failure, α-Syn aggregation, and the impairment of proteasome systems in the neurodegenerative process of PD [132,133].

Transcriptional dysfunction of several genes has been associated with familial PD [134,135]. These defects include changes in the expression of α-Syn, Parkin, leucine-rich repeat kinase 2 (LRRK2), PTEN-induced putative kinase 1 (PINK1), and DJ1 [136]. From this group, the major autosomal dominant PD-related gene is *LRRK2*, which is considered to be the most common genetic cause of familial and sporadic PD [137]. This genetic deficiency has been demonstrated to cause α-Syn accumulation and autophagy stress and the disturbances in mitochondrial dynamics (fusion/fission), mitochondrial membrane potential, and mtDNA [138]. The loss of Parkin function could harm mitochondrial biogenesis and conduct cell death [128,139]. PINK1 accumulates on dysfunctional mitochondria, and its kinase activity is required for Parkin translocation to mitochondria and the induction of the mitophagy [140,141]. Additionally, the loss of function mutations of the *DJ-1* gene has been associated with the recessive early-onset familial PD and late-onset sporadic PD [142]. Changes in *DJ-1* expression resulted in defective complex I function, fragmented mitochondria, uncoupling, Ca^2+^ disturbances, and oxidative damage [143]. Altogether, these studies showed that mutations in these genes affect mitochondrial function, contributing to PD’s neurodegeneration [144].

### 3.1. PD and ROS

Accumulative damage by an uncontrolled ROS production in the brain contributes significantly to the pathogenesis of PD [145,146]. In physiological conditions, ROS are produced from several sources including complexes I and III of the ETC located in the mitochondria inner membrane and can also be produced by NADPH oxidase induced by Ca^+2^ influx [8,147]. In a pathological condition such as PD, oxidative damage and mitochondrial impairment contribute to the cascade of events leading to the degeneration of dopaminergic neurons, considered one of the hallmarks of this disease [148] (Figure 2). More importantly, the experimental data suggest that ROS overproduction is a significant contributor to dopaminergic neuronal loss in PD [149] (Figure 2).

In homeostatic conditions, GSH plays a fundamental role in reducing high levels of ROS and minimizing oxidative damage in the brain [150]. In PD, the extended loss of nigral GSH is the most distinctive change that happens in the earliest stage of this progressive disease [151]. Analysis of postmortem brain from PD patients showed a decrease in the amount of GSH in the Sustantia nigra (SN) compared to the controls [152]. Loss of GSH in the SN results in diminished levels of mitochondrial complex I activity [153]. In the same context, accumulation of iron ions (Fe^3+)^ and ferrous iron (Fe^2+^) can easily react with radical superoxide (O_2−_) and hydrogen peroxide (H_2_O_2_), producing a highly reactive hydroxyl free radical that, together with dopamine oxidation, can trigger neurotoxicity observed in PD [149].

### 3.2. PD and Mitochondrial Impairment

Oxidative stress is considered one of the critical factors in the etiopathology of PD [145,148]. A significant amount of ROS levels are produced by the inhibition of complex I of the mitochondrial respiratory chain, and this alteration is present in PD patients [154]. During PD, increased ROS production could affect mitochondrial function, inducing calcium uptake by oxidizing thiol groups in protein channels, which later contributes to neuronal death [155]. Dopamine oxidation could affect mitochondrial function by producing an increase in ROS levels [156,157] (Figure 2). Primary or toxin-induced respiratory chain complex dysfunction in PD is tightly linked with ROS generation [145]. During PD, dopaminergic neurons showed a ROS accumulation contributing to mitochondrial impairment, and the inflammatory response shown in PD [158]. How important is mitochondrial impairment for the pathogenesis and neurodegeneration in PD will be a matter of discussion in the next sections.

#### 3.2.1. Mitochondrial Bioenergetics Defects

The first association between PD and mitochondrial dysfunction was suggested in the late 1970s by using the neurotoxic compound MPTP (1-methyl-4-phenyl-1,2,3,6-tetrahydropyridine), a by-product accidentally generated during the synthesis of a meperidine analogue [159]. Treatment with MPTP caused parkinsonian-like symptoms in intravenous drug users [159,160]. When MPTP crosses the blood–brain barrier, this compound is bio-transformed into its toxic form 1-methyl-4-phenylpyridinium (MPP+) by glial monoamine oxidase (MAO) [161]. MPP+ specifically interferes with the activity of mitochondrial respiratory chain complex I (NADH: Ubiquinone oxidoreductase) in dopaminergic neurons (DA), causing selective neurodegeneration in the substantia nigra of both human and mouse models [159,162]. As a consequence of the mitochondrial respiratory complex I defects, the ATP production decreases while the generation of ROS and nitrogen species is increased [163,164]. These actions ultimately lead to neuronal cell death by activation of pro-apoptotic Bcl-2 family members, p53, JNK, and caspases as well as inflammation [165,166].

Furthermore, mitochondrial respiratory complex I activity was significantly reduced in the SN of PD patients [167,168,169]. Additionally, the higher number of mitochondrial DNA (mtDNA) encoded subunits required for complex I assembly indicates that this complex is more likely to be affected by pathogenic mtDNA mutations [167,170]. However, the specific role of complex I deficiency in PD neuronal loss remains elusive [171,172]. In addition, a wide number of clinical trials showed a relation between PD and specific mtDNA-specific mutations. Interestingly, the mitochondrial transcription factor A protein (Tfam), whose expression sequence is part of the promoters within the D-loop region of mtDNA, have been implicated in PD [173]. Studies in Tfam (−/−) knock-out mice showed a decrease in mtDNA expression, respiratory rate deficiency, neuronal death, and the impairment of motor functions in midbrain dopaminergic neurons [174].

#### 3.2.2. Mitophagy

Mitophagy, which is known as the selective degradation of mitochondria through autophagy, is an elemental mechanism for mitochondrial homeostasis [175,176]. This process is particularly essential in post-mitotic and slow-dividing cells (like neurons) as it promotes the renewal of mitochondria, preventing the accumulation of dysfunctional organelles [177]. Alteration of mitophagy leads to the progressive accumulation of defective mitochondria, leading to neurodegeneration and synaptic dysfunction [178,179] (Figure 2). Clinically, the majority of PD cases are late-onset and sporadically caused by a combination of genetic and environmental factors; up to 10% have origin in monogenic forms of the disease [137]. Interestingly, the identification of familial autosomal recessive forms of early-onset PD caused by mutations in *PINK150* and *PARKIN51* genes has been critical in implicating dysfunctional mitophagy in the PD pathogenesis [178,180,181].

Under physiological conditions, PINK1 localizes with the mitochondria and is quickly translocated toward the inner membrane (IM) to be cleaved and deactivated by IM protease presenilins-associated rhomboid-like protein (PARL) [182]. PINK1 inhibits the fusion of damaged mitochondria, which may serve the purpose of preventing the contamination of healthy mitochondria by those that are damaged [182,183]. In addition to the adverse effects against mitochondrial function and dynamics, PINK1 and Parkin participate in the mitochondrial biogenesis process through an indirect interaction with the cofactor PGC-1α [139,184]. PGC-1α (encoded by *PPARGC1A* gene) was discovered as the co-regulator of PPARγ, a transcriptional factor that is considered a master regulator of mitochondrial biogenesis [184,185]. The loss of PINK1/Parkin activity reduces clearance and the consequent build-up of the Parkin interacting substrate (PARIS/ZNF746), a transcriptional repressor of PGC-1α [186]. Interestingly, Murata and colleagues found that Nrf2 is also a transcription factor for the *PINK1* gene via activation of the ARE sequence in the PINK1 promoter (Figure 1). This Nrf2-PINK1 signaling participates in cell survival and in the maintenance of mitochondrial homeostasis through several mechanisms such as the removal of damaged mitochondria and reduction of ROS [187].

Current studies have indicated a possible explanation of how PINK1 may regulate parkin-mediated mitophagy and how PD-associated PINK1 and parkin mutations result in defective mitophagy [188]. For example, experiments using SH-SY5Y human neuroblastoma cells that were stable knockdown for PINK1 showed an enhancement of the mitophagy process by increasing oxidative stress [189]. In the same context, the overexpression of PINK1 stabilized mitochondrial networking and function in SH-SY5Y neuroblastoma cells [189]. Complementary evidence also demonstrates that Parkin enhanced this protective mitophagy response, indicating that PINK1 and Parkin may cooperate to maintain mitochondrial homeostasis [189,190]. In addition, the suppression or knock-out of Parkin in Drosophila, zebrafish, mice, or human patient cells leads to severe mitochondrial dysfunction including decreased ATP production, mitochondrial depolarization, and altered mitochondrial morphology [191,192]. In particular, one study showed evidence that Parkin interacts with mitochondrial Stomatin-like protein 2 (SLP-2), which is a protein required for the assembly of mitochondrial respiratory chain complexes [193]. These studies suggest that mutations in both Parkin or PINK1 may alter mitochondrial turnover, resulting in the accumulation of abnormal mitochondria that contribute to PD [194,195].

### 3.3. Nrf2 Activation Prevents Neurodegeneration in PD

Accumulative studies suggest that the use of natural or synthetic compounds that activate the Nrf2 pathway reduce the negative consequences of the oxidative damage and mitochondrial dysfunction present in NDs [196]. Dimethyl fumarate (DMF) is a methyl ester of fumaric acid, and its neuroprotective actions are conducted by modulating the Nrf2 pathway [197]. DMF acts as an Nrf2 activator with the capacity to stimulate a cellular defense to protect neurons from ROS damage [198]. Similarly, other groups have studied the effect of Kolaviron, a mixture of bioflavonoids with neuroprotective effects on the microglia activation [199]. BV2 microglia and HT22 hippocampal neuron co-culture treated with this compound showed a reduction in PGE2/COX-2, and NO/iNOS, and an increase in HO-1 levels promoting an antioxidant environment in these cells [199].

Strong evidence suggests that the activation of Nrf2 could be a therapeutic target against PD and other neurodegenerative diseases such as AD and HD [200,201,202,203]. At least two relevant facts support the importance of the Nrf2 pathway on PD: first, the *DJ-1/PARK7* gene, whose deficiency is associated with autonomic recessive parkinsonism, sequesters the Nrf2 inhibitor Keap1 and leads to an increase in Nrf2 activity [204]. *DJ-1*-deficient patients showed a reduced expression of Nrf2-dependent genes such as *NQO1* and *GST*, increasing the oxidative damage [205,206,207]. The second fact is the association between PD and the polymorphisms of several genes regulated by Nrf2 [16,208,209]. This association was suggested by studies of the microchip analysis of tissues samples of PD patients [16]. Brain PD samples showed a decrease in expression of 31 genes that contained the ARE-sequences in the promoter with an increase in the expression of Nrf2 [16]. Interestingly, other studies have suggested that the Nrf2 pathway is highly activated in PD [210,211]. Moreover, neuronal cells such as astrocytes, endothelial cells, and dopaminergic neurons showed an increase in NQO1 levels in PD post-mortem brain samples [203]

GSK-3β has been implicated in the regulation of several physiological responses by phosphorylating a wide range of nuclear and cytoplasmic proteins [212,213]. Additionally, GSK-3β participates in several processes related to mitochondrial function such as biogenesis, bioenergetics, motility, and permeability [214]. In the PD context, an association between PD and two single nucleotide polymorphisms located in the GSK-3β promoter region, (nt-171 to +29), rs334558 (−50 C/T), and intronic single nucleotide polymorphism, rs6438552, has been reported [215]. These interactions affect GSK-3β function by increasing its expression and activity [215]. In addition, Rojo and colleagues identified GSK-3β as a fundamental element in the downregulation of the antioxidant defense elicited by Nrf2 after oxidant injury [42]. In this case, the GSK-3β inhibition resulted in increased Nrf2 nuclear levels and the expression of antioxidant genes that prevented the oxidation induced by H_2_O_2_, 6-hydroxydopamine (6-OHDA), and MPP+ [42,45].

Complementary studies showed that mice neuronal cultures co-transfected with Nrf2 and α-synuclein showed an increase in proteasomal activity, conducting a reduction in the neuronal load of α-synuclein [216]. Interestingly, pioneer studies from Jakel et al. showed that the transplantation of astrocytes overexpressing Nrf2 into the striatum of a Nrf2 (−/−) PD mice model induced by 6-OHDA showed neuroprotection and reduced oxidative damage [217]. Importantly, Nrf2 activation reduced oxidative damage and neuroinflammation of wild type mice treated with MPTP [218]. Additionally, in Nrf2 (−/−), knock out mice submitted to MPTP showed a decrease in Nrf2 activity, which accentuates the PD pathological phenotype [219].

Despite the fact that Nrf2 has not been evaluated for clinical trials in PD, epidemiological evidence indicates that the intake of compounds that activate this pathway such as vitamins C and E are associated with a significant decline in PD risk [220]. Additionally, the treatment during four weeks with N-acetyl cysteine, which activates the Nrf2 pathway, showed better scores on the Unified Parkinson Disease Rating Scale and increased peripheral markers of antioxidant activity [221].

Further studies have shown that the Nrf2 activation by sulforaphane (SFN) induced the expression of antioxidant enzymes, reduced ROS levels, increased mitochondrial biogenesis, and prevented dopaminergic neuronal loss in MPTP-treated mice [222]. SFN is a natural organic isothiocyanate compound isolated from the cruciferous family of vegetables, which includes cauliflower, brussels sprouts, broccoli, white cabbage, and red cabbage [223,224]. Increasing evidence has shown that this compound could act on different targets and modulate antioxidant response in neuronal cells [225,226]. The lipophilic nature and molecular size of SFN make it suitable for passive absorption by neuronal cells, where SFN is attached with GSH by glutathione S-transferase (GST), which leads to the maintenance of a concentration gradient by a continuous uptake of SFN [227].

In this context, studies have reported that neurons treated with SFN altered Keap1 function mainly in the cysteines (Cys-77, Cys-226, Cys-249, Cys-257, Cys-489, Cys-513, Cys-518, and Cys-583) of the Kelch domain of human Keap1 [228]. Furthermore, SFN suppressed the GSK-3β activity and increased the nuclear translocation of Nrf2 [229]. More importantly, SFN pre-treatment decreased membrane damage and DNA fragmentation, and ROS generation caused 6-OHDA treatment on rat nigrostriatal cultures [230]. These positive effects were mediated by Nrf2 activation by increasing enzyme levels of NQO1 [230,231]. Likewise, SFN protected against 6-OHDA treatment in cultures of rat nigrostriatal neurons [232] and human neuroblastoma SH-SY5Y cells [233].

## 4. Neurodegeneration in AD

AD is the most common type of dementia in the elderly population and is characterized by memory loss and selective neuronal death [234]. AD brain develops two distinguished pathological characteristics: the senile plaques formed by extracellular deposits of the β-amyloid peptide Aβ (1–40), Aβ (1–42) and the neurofibrillary tangles (NFTs) made of intraneuronal aggregations of hyperphosphorylated tau protein [235]. Accumulative evidence has shown extensive mitochondria abnormalities in AD patient’s brains [236]. Additionally, mitochondrial impairment has been established as an early hallmark in the genesis and progression of this disease [236,237]. In addition, the GSK-3β dysregulation is related to the pathogenesis of AD [234]. Elevated GSK-3β activity has been observed in the brains of AD patients [238]. Several studies have proposed a molecular relation between GSK-3β, Aβ, and tau in AD pathogenesis [239]. It has been demonstrated that Aβ activates GSK-3β, which in addition, hyperphosphorylates tau protein (Figure 2) [240]. Mice that overexpress GSK-3β in the forebrain (Tet/GSK-3β mice) exhibit hyperphosphorylation of tau, followed by its accumulation in hippocampal neurons [241].

### 4.1. ROS in AD

It is well accepted that oxidative stress plays a significant role in the pathogenesis of AD [7,242,243]. In this context, Aβ (1–42) mediates part of its harmful effect against neurons by inducing mitochondrial stress and ROS increase in AD patients [244]. In addition, increase of ROS levels promotes Aβ deposition and the loss of synapses in hippocampal neurons [245]. Studies using live imaging showed an active ROS production followed by neuronal death in the proximity of amyloid plaques in the APP/PS1 double transgenic mice that exhibited elevated production of β-amyloid [246]. Several findings have demonstrated that lipid peroxidation is highly increased in AD [242]. Lipid peroxidation is a process in which ROS attack lipids through a mechanism of free radical chain reaction, generating lipid peroxidation products harmful for neuronal function such as 4-hydroxy-2-nonenal (HNE), which is one of the major products of lipid peroxidation that also contributes to Aβ depositions in AD [242,247]. Protein oxidation by ROS also plays a significant role in the modification of key enzyme activities critical to neuron and glial functions such as glutamine synthetase and creatine kinase, which are significantly reduced in AD brains [248]. In addition, increased levels of 3-nitrotyrosine, a marker for peroxynitrite-mediated protein nitration, have also been found in brain regions of AD patients compared to control samples [249]. Thus, these irreversible protein modifications that include protein nitration and HNE could compromise enzymatic activity and alter energy metabolism in AD.

### 4.2. AD and Mitochondrial Impairment

Interestingly, a strong connection has been shown between mitochondrial dysfunction and oxidative stress in the progression of AD [250]. These defects represent a critical aspect to explain how oxidative damage and synaptic dysfunction contribute to the early stages of AD [251,252,253]. Oxidative damage is considered an early effect in AD, as recent studies have shown that its onset is frequently preceded by an intermediate phase known as mild cognitive impairment (MCI) when there is no accumulation of senile plaques and NFTs [254]. In addition, increased levels of Aβ could accelerate ROS production by directly binding to mitochondrial membranes, thus affecting mitochondrial dynamics and function, which leads to abnormal energy metabolism and synaptic loss [255].

#### 4.2.1. Mitochondrial Dynamics Defects

Mitochondrial dysfunction is an essential and early feature of AD, and almost all aspects of mitochondrial biology have been reported to be affected during this disease [256]. Mitochondria suffer from continuous fission and fusion (mitochondrial dynamics) events that regulate their morphology and distribution [257]. These processes are coordinated by mitochondrial fission proteins such as dynamin-like protein 1 (DLP1), Fis1 [258], and mitochondrial fusion proteins like mitofusin 1 (Mfn1), mitofusin 2 (Mfn2), and Optic atrophy 1 (OPA1) [259]. Most DLP1 expression is present in the cytoplasm. Nevertheless, during fission, DLP1 moves toward the mitochondrial surface and arises as a dot formation [259]. Fis1, Mfn1, and Mfn2 are mitochondrial transmembrane proteins placed on the outer mitochondrial membrane [257], while OPA1 is located on the inner mitochondrial membrane [260]. In AD, alterations of mitochondrial dynamics have been demonstrated, showing significant changes in the expression of almost all mitochondrial fission and fusion proteins in the brain of AD patients [256]. Mitochondrial DLP1, the critical fraction for mitochondrial fission, increases in the brain with AD [261]. Consistently, the phosphorylation of DLP1 in Ser616 and S-nitrosylation of DLP1 enables DLP1 translocation to mitochondria activating the GTPase activity of DLP1, which induces mitochondrial fission [262]. In addition, structural damage to mitochondria such as the presence of broken ridges and almost complete loss of internal structure represents a frequent sign in AD brains [263]. Furthermore, Wang and colleagues also demonstrated that Aβ production causes an imbalance of mitochondrial fission/fusion, resulting in mitochondrial fragmentation and abnormal distribution [264]. Furthermore, Aβ treatment causes oxidative stress and induces mitochondrial fragmentation through decreased expression of mitofusin-2 (Mfn2) by activating cyclin-dependent cyclin 5 (Cdk5)-mediated phosphorylation of peroxidase 2 (Prx2) [265]. In addition, Aβ mediates the phosphorylation of dynamin-related protein 1 (Drp1) through AKT stimulation, causing an increase in mitochondrial fission and neuronal death [264]. Importantly, significant changes in the size and number of mitochondria have also been found in susceptible AD neurons [266].

#### 4.2.2. Mitochondrial Bioenergetics Defects

It has been established that damaged mitochondria are less efficient in ATP production, but more capable in the generation of ROS, a fact that represents a key source of the oxidative imbalance observed in AD [266]. In this context, exciting studies suggest that mitochondrial damage resulting from increased ROS production could be an essential contributor to the early stages of AD before reaching the onset of clinical symptoms and the appearance of the tau and Aβ pathology [267].

Complementary studies have also shown that Aβ induces ROS generation and impaired calcium homeostasis, leading to mitochondrial dysfunction [236,255]. For example, the overexpression of the mutant beta-amyloid precursor (APP) protein in the hippocampal cell line HT22 results in defective mitochondrial dynamics, changing the structure and function of these organelles [268]. APP can accumulate in the mitochondrial import channels of AD brains and cause mitochondrial dysfunction [269]. Additionally, Aβ can directly affect mitochondrial function, inhibiting some key enzyme activities [270]. For example, Lustbader et al. showed that alcohol dehydrogenase (ABAD) interacts with Aβ and mediates Aβ-induced apoptosis and ROS production in neurons [271]. Other studies have also shown an increase in the levels of voltage-dependent anion-selective channel 1 (VDAC1), Aβ, and hyperphosphorylated tau in AD brain samples [268].

Furthermore, the disruption in glucose metabolism is associated with early mitochondrial defects observed in several studies made in animal models and AD patients [255,272,273]. These studies showed a significant reduction in glucose metabolism consistent with a first feature observed during AD [274,275]. In addition, the examination of the mitochondrial bioenergetics profile in fibroblasts from late-onset AD (LOAD) and age-healthy control match patients demonstrated that the cells from LOAD changed their metabolic activity from the mitochondrial oxidative phosphorylation system (OXPHOS) to glycolysis generating mitochondrial depolarization in LOAD cells [276]. Finally, brain mitochondria fractioned from triple transgenic AD model mice (3xTg-AD) showed mitochondrial depolarization, a decrease in ATP production, and respiratory rate failure [277].

#### 4.2.3. Mitochondrial Transport Defects

Mitochondrial transport is a process that contributes to the proper organelle distribution through the neurons [278]. Recent studies have shown defects in axonal mitochondrial transport, which precedes to the accumulation of toxic protein in AD, altering axonal integrity and synaptic performance [251,279,280]. In fact, the relation between mitochondrial transport and synaptic activity has been investigated when a loss of function of syntabulin, an adaptor protein that mediates presynaptic mitochondrial motility, affects synaptic plasticity [281,282]. While the exact molecular mechanism in the mitochondrial transport inhibition seen in AD is still a matter of research, a disturbance in mitochondrial motility is highly connected with increased levels of both Aβ and hyperphosphorylated tau and oxidative stress [280,283]. For example, Reddy’s group showed alterations in anterograde mitochondrial movement with an increased fission, and mitochondrial depolarization in primary neurons obtained from the Tg2576 mice expressing mutant human APP protein [284].

#### 4.2.4. AD, Tau Pathology, and Mitochondrial Dysfunction

Tau is a neuronal protein that exhibits different post-translational modifications including phosphorylation, glycosylation, acetylation, nitration, methylation, prolyl isomerization, ubiquitylation, sumoylation, and glycation [285,286]. In AD brains, tau is pathologically modified, which causes its separation from microtubule structures, leading to intraneuronal aggregation and the formation of NFTs [287]. Defects in tau could accelerate neurotoxicity or become neurons more susceptible to different stressors including calcium dysregulation, oxidative stress, inflammation, and mitochondrial failure [288]. In this context, transgenic expression of truncated human tau, which is a relevant tau modification in AD [289], affects mitochondrial distribution and decreases neuronal viability under conditions of exogenous oxidative stress [290]. Furthermore, it has been established that mutated tau expression at P301L affects mitochondrial dynamics, respiratory activity, decreases ATP levels, and induces mitochondrial depolarization in SY5Y cells exposed to hydrogen peroxide [291]. In this context, further studies have shown that an important proportion of 20–22 kDa N-terminal tau fragments (NH2hTau) are found preferentially in the AD hippocampus and frontal cortex mitochondria [292,293]. This tau fragment is connected with neurofibrillary degeneration, synaptic damage, and mitochondrial impairment in the AD brain [292]. Interestingly, it has been suggested that this N-terminal tau truncation contributes to disease progression and represents a key step in the toxic cascade that leads to neuronal death [294].

In this context, recent studies have shown that a reduced tau expression could enhance neuronal and mitochondrial function [295]. For example, Lopes et al. showed that genetic reduction of tau prevented impaired working memory, loss of dendritic spine, and synaptic failure induced in a mouse exposed to chronic stress [295]. More importantly, Jara et al. showed that tau knock-out mice presented less oxidative damage, better recognition memory and attention span, and better mitochondrial bioenergetics compared to wild-type mice that expressed physiological tau levels [296]. In addition, reduced tau expression prevented oxidative damage in hippocampal cells and activated the pathways necessary to protect mitochondrial health such as the Nrf-2 and PGC-α pathways [1,296]. Therefore, these studies suggest that stress-induced neuronal damage and cognitive decline depend on an interaction between tau and mitochondria, which could later affect memory and cognition.

In addition, other pathological tau modifications like caspase cleavage play a pivotal role in the pathogenesis of AD [297,298,299]. Tau could be a substrate for caspase-3 and is cleaved at Asp-421, which is located at the protein’s carboxyl-terminus [300]. This cleavage event results in a highly fibrogenic protein form that rapidly assembles into tau filaments in vitro than that of wild-type tau [297]. Different studies made in cell culture models provided evidence that Asp-421 cleaved tau is toxic to neurons [294,300,301].

More importantly, we have carried out several studies investigating the effects of caspase 3-cleaved tau against neuronal viability induced by mitochondrial failure [301,302,303,304]. We showed that immortalized cortical neurons expressing caspase-3-cleaved tau presented severely fragmented mitochondria, high ROS levels, and mitochondrial depolarization when cells were exposed to calcium overload stress [301]. In addition, other studies using a pseudophosphorylated form of tau (PHF-1), a component of NFTs and caspase 3-cleaved tau, showed that these pathological forms contribute differently to mitochondrial impairment in neurons [304]. Neurons expressing the tau pseudophosphorylated form (PHF-1) affected mitochondrial function (potential and ROS production) in mature neurons; in contrast, hippocampal neurons expressing truncated tau affected mitochondrial function regardless of the age of neuronal culture [304]. These results support the hypothesis that truncated tau at Asp-421 is an early contributor to mitochondrial impairment in the AD brain, and these effects could be relevant for the disease progression [297,301].

Interestingly, new findings suggest that pathological forms of tau and specially truncated tau by caspase-3 may also affect mitochondrial transport in hippocampal neurons [303,304,305]. Truncated tau expression reduced the number of moving mitochondrial elements compared with hippocampal neurons expressing full-length tau [304]. In addition, we studied the molecular mechanism involved in the mitochondrial transport failure induced by truncated tau [305]. Interestingly, truncated tau expression affects the expression of TRAK2 protein, which is a critical mitochondrial-associated transport accessory protein [306]. Truncated tau augmented the accumulation of mitochondria in the soma, reducing the mitochondrial population in neuronal processes including the axon [305]. Interestingly, truncated tau reduced TRAK2 expression and increased its interaction with mitochondria compared to cells expressing full-length tau [305]. These novel findings indicate that caspase-cleaved tau may affect mitochondrial transport through increasing TRAK2-mitochondrial binding and also affecting ATP production [305]. Together, these findings reveal the importance of tau participation in the mitochondrial transport in neurons, an event that may contribute to the synaptic failure observed in AD.

### 4.3. Nrf2 Activation Reduces Tau Pathology in AD

Several studies under the cell-based and in silico high-throughput screens have identified Nrf2-activating natural compounds that control the expression of genes linked with autophagy response and nerve growth factor signaling in the CNS [20]. These wide arrays of functions developed by Nrf2 might also ameliorate disease progression [307]. In this context, evidence obtained by Jo et al. showed that activating the Nrf2 pathway reduced the abnormal accumulation of hyperphosphorylated tau by inducing the expression of autophagy adapter protein NDP52 in neurons [308]. The NDP52 protein presents three ARE in its promoter region and its expression is strongly induced by Nrf2 activation, facilitating the elimination of hyperphosphorylated tau [308]. The polyubiquitination binding protein, p62, can also modulate the Nrf2 activity through Keap1, which targets its degradation by autophagy [108]. Other studies have shown that Nrf2-mediated induction of p62 plays a critical role in the TLR4-driven aggresome formation and the autophagy degradation to maintain host protection [306]. These studies are related to early evidence that showed significantly decreased levels of p62 in the frontal cortex of AD patients [309]. On the other hand, mRNA and p62 protein levels as well as Nrf2 target genes were shown to be augmented in the cortex of AD brains [310,311]. Furthermore, Aβ injection into rat hippocampus induced an increase in the expression levels of LC3-II, beclin1 (both autophagy protein markers), and Keap1 while p62 and Nrf2 levels were decreased in the hippocampus and cortex of these animals [312].

Cuadrado et al. investigated the effect of the expression of pathological forms of tau like P301L on Nrf2/ARE pathway activation in the Mefs cell line (Keap1 −/−) and Nrf2 (−/−) knock out mice [73]. P301L expression was made with the use of recombinant Adenovirus-Associated-Virus vectors (AVV hT-AU^p301L^) [73]. Furthermore, the Nrf2 (−/−) mice model expressing AVV-hT-AU^p301L^ showed an increase in the levels of tau hyperphosphorylated, and this effect was prevented in Nrf2 (−/−) mice treated with DMF [73]. Treatment with DMF induces the Nrf2 transcriptional activation through KEAP1-dependent and-independent mechanisms both in vitro *and* in vivo model [313]. These studies suggest that the impairment of the Nrf2 pathway could be involved in the modulation of tau pathology and the formation of neurofibrillary tangles.

### 4.4. Nrf2 Activation Prevents Neurodegeneration and Mitochondrial Failure in AD

Accumulative evidence has suggested an essential role of the Nrf2 pathway as a useful therapeutic strategy to ameliorate neurodegenerative changes present in AD [314]. Ramsey and colleagues demonstrated a significant expression of Nrf2 in the nucleus of primary hippocampal neurons while the examination of AD brain samples showed a major Nrf2 cytoplasmic presence compared with the control tissue samples [315]. This reduced presence of Nrf2 in the nuclei of hippocampal neurons during AD suggests that Nrf2 is not performing the activities observed in control individuals of the same age [315]. This deficit in nuclear Nrf2 presence is not the result of a generalized loss of the total cytosolic protein, but may reflect an altered nuclear trafficking, as seen with other transcription factors [316,317].

Furthermore, studies made in Nrf2 (−/−) mice’s hippocampus exhibited higher levels of oxidative damage and pro-inflammatory factors compared with wild type mice with endogenous Nrf2 expression [318]. In this context, memory and learning performance was severely affected in Nrf2 (−/−) mice as early as six months and before the appearance of amyloid plaques and fibrillary aggregates formed by tau [319,320].

Current evidence shows that increased antioxidant activity reduces the risk of neurodegenerative diseases [74,321]. Nrf2 appears to be a promising target because several natural Nrf2 activators have shown positive effects in vitro and in vivo models of study for age-related neurodegenerative diseases [20].

The use of natural compounds with mitochondrial boost actions has also been studied to treat neurodegenerative diseases like AD [322,323]. Various vegetable components like SFN from broccoli or spice ingredients like S-allyl-l-cysteine (SAC) from garlic are known to activate the Nrf2 pathway [324]. Complementary studies have shown that the pharmacological activation of Nrf2 with SFN or through the use of lentiviruses for the activation of Nrf2 increases the expression of antioxidant genes regulated by ARE, improving long-term memory defects induced by Aβ peptide both in vitro and in vivo [314,325]. Further studies have examined the effect of elevated expression of the proteasome (i.e., 26S) in cytoprotection by SFN in murine neuroblastoma cells [72]. These studies propose that the proteasome system’s upregulation leads to the cytoprotective effects induced by SFN, in opposition to oxidative stress [72]. Another study from Park et al. reported that the SFN prevents neuronal cells from Aβ_42_ treatment and also reduces proteasome activity [326]. More importantly, complementary studies showed that SFN improved neurobehavioral deficits and decreased the load of Aβ in the AD transgenic mice model (APP/PS1) where treatment with SFN increased p75 neurotrophin receptor levels by reducing histone deacetylase 1 and 3 expressions [327]. Furthermore, SFN treatment of neuroblastoma cells upregulated autophagy genes, which were suppressed after Nrf2 expression was knocked out, suggesting a relevant role of Nrf2 in neuroprotection against the toxicity of Aβ [69].

Besides SFN, other natural products have shown positive effects in reducing the oxidative damage and mitochondrial injury during AD [328,329]. For example, a compound known as curcumin exerts neuroprotection in AD [328]. Curcumin is a natural phytochemical compound isolated from the rhizome of the plant named turmeric, which has been shown to have multiple positive effects for neuronal function including anti-inflammatory, antioxidant, and anti-protein-aggregate properties [330]. Curcumin exerts neuroprotection for its capacity to scavenge free radicals such as ROS and reactive nitrogen species (RNS) [331]. In addition, curcumin exhibits protection to mitochondria in the prevention of lipid peroxidation and protein carbonylation [332]. Furthermore, curcumin-derived product (CNB-001) protects mitochondrial dysfunction by its capacity to maintain mitochondrial membrane potential (ΔΨm) and mitochondrial respiratory complex activity under physiological levels [333,334].

DMF is another product that has been shown to have positive outcomes against neurodegeneration in PD and AD [21,335]. DMF has been approved by federal drug administration (FDA) mainly for the treatment of multiple sclerosis (MS) and is also considered as a promising agent for the treatment of AD [336,337]. DMF has protective effects against oxidative stress by increasing the production of the NQO1 enzyme, which exerts a neuroprotective function in hippocampal pyramidal neurons in AD patients [338]. In addition, Kume and colleagues used primary striatal cells that showed that DMF treatment reduced oxidative stress in striatal cells exposed to sodium nitroprusside [339]. Interestingly, these positive effects were produced because DMF treatment induced an increased activity of *HO-1* and other ARE-related to neuroprotective genes [339]. In addition, DMF induces mitochondrial biogenesis mainly through the activation of Nrf2, as shown by one in vivo human study [340]. Furthermore, treatment with DMF alleviated tauopathy, decreased GSK-3β activity, and promoted neuronal viability by increasing brain-derived neurotrophic factor (BDNF) expression and reduced inflammatory processes in the hippocampus of Nrf2 −/− mice that stereotaxically expressed the human *TAUP301L* gene, which is a tau modification responsible for frontotemporal dementia [73].

Simultaneously, it is possible to exert negative regulation of the Nrf2 pathway by several kinases constitutively activated or over-expressed in pathological conditions such as chronic inflammation and AD [341,342]. As above-mentioned, GSK-3β can induce the degradation of Nrf2 by the proteasome in a Keap1-independent manner [341]. These are exciting findings because GSK-3β expression is upregulated in the hippocampus of AD patients [343], post-synaptosomal fractions from AD brains [342], and in AD circulating peripheral lymphocytes [343].

Current reports have elucidated the possible links between GSK-3β and Nrf2 in AD pathology, suggesting new neuroprotective therapies [344]. For example, Rojo and colleagues showed that the inhibition of GSK-3β with lithium administration increased the transcriptional activity of Nrf2 in N2A cells [42]. In addition, cortical extracts obtained from the AD mouse model with GSK-3β suppression showed an increase in nuclear Nrf2 localization, an increase in glutathione-S transferase (GST) levels, reduced oxidative damage, and a decrease in tau hyperphosphorylation levels [345]. Additionally, pyrrolidine dithiocarbonate (PDTC), a small molecule with antioxidant properties that inhibits GSK-3β, also showed Nrf2-ARE stimulation and a significant improvement of the cognitive decline present in APP/PS1 mice [346]. Furthermore, the PI3K/Akt pathway, which downregulates GSK-3β and promotes Nrf2 activity [347], was reduced in the brain of patients with AD [348]. Furthermore, the antioxidant puerarin, described to present antihypertensive, antiarrhythmic, antioxidant, anti-apoptotic, and neuroprotective properties [349,350], caused augmented PI3K/Akt stimulation, reduced GSK-3β activity, increased Nrf2 pathway activation, and *HO-1* gene expression in APP/PS1 mice hippocampus, with further cognitive improvement [350].

Notably, a new AD therapy that simultaneously inhibits GSK-3β and increases Nrf2 activity has been proposed [344]. Interestingly, therapeutic Nrf2 activation with 5A-T compounds was independent of GSK-3β inhibition in SH-SY5Y human neuroblastoma cells treated with the mitochondrial toxins oligomycin/rotenone [344]. Nowadays, the number of natural and synthetic Nrf2 activators that can be used for a better understanding of the positive effects that Nrf2 may have on AD pathology are still increasing.

## 5. Conclusions

In this review, we discussed several studies that highlighted the protective role of the Nrf2 pathway against PD and AD. Moreover, upregulation of Nrf2-driven antioxidant enzymes decreased neurodegeneration and mitochondrial dysfunction both in vitro and in vivo models of PD and AD (Figure 2). In this context, we considered that the Nrf2-ARE pathway could be a valid therapeutic target to ameliorate the progression and pathological signs present in these disorders.

Evidence from animal models indicate that a pharmacologic therapy to slow the progression of neurodegenerative diseases might be achieved through the activation of Nrf2. Nrf2-activating compounds are already FDA approved for the treatment in other diseases such as multiple sclerosis, and clinical testing is underway in Friedrich’s ataxia [350]. In addition, Nrf2 activators can also prevent mitochondrial impairment, which is related to synaptic dysfunction in PD and AD. Therefore, activation of the Nrf2 pathway could have positive effects by decreasing oxidative damage and also reducing the cognitive impairment present in neurodegenerative disorders like PD and AD.

## Figures and Tables

**Figure 1 antioxidants-10-01069-f001:**
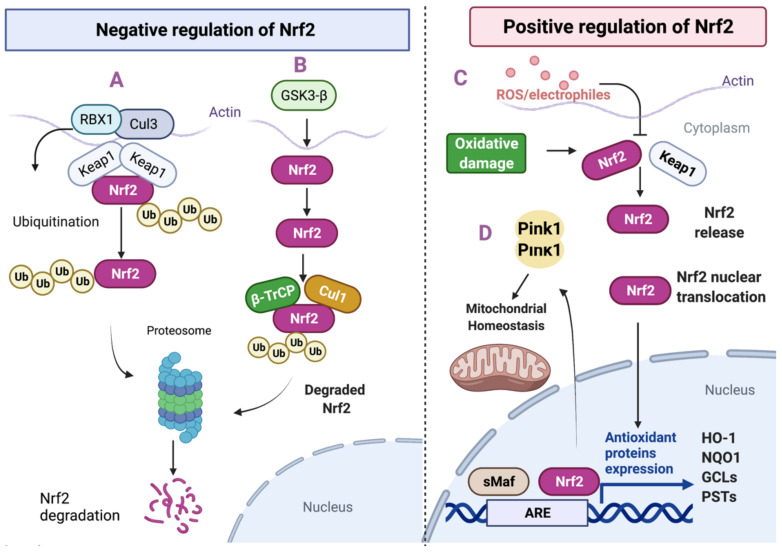
Regulation of the Nrf2 pathway in neuronal cells. (**A**) Negative regulation of Nrf2. Under basal conditions, Nrf2 expression is maintained at low levels through the proteasome activity. Nrf2 is sequestered by Keap1 in the cytosol leading to ubiquitination through the formation of the Keap1–Cul3-Rbx1 complex, which induces the Nrf2 proteasomal degradation. (**B**) Negative regulation of Nrf2 by GSK-3β: Nrf2 can also be conducted to degradation by GSK-3β. GSK-3 phosphorylates Nrf2 to create a recognition motif for the E3 ligase adapter β-TrCP. GSK-3/β-TrCP leads to Keap1-independent ubiquitin-proteasome degradation of Nrf2. (**C**) Positive regulation of Nrf2: in the presence of high ROS levels, Nrf2 is released from Keap1 binding and is translocated to the nucleus and binds to ARE gene sequences, which allows for the activation of antioxidant genes such as HO-1 and NQO1. (**D**) Mitochondrial homeostasis: PINK1 expression is positively regulated by Nrf2 and promotes mitochondrial homeostasis through several mechanisms such as the removal of damaged mitochondria.

**Figure 2 antioxidants-10-01069-f002:**
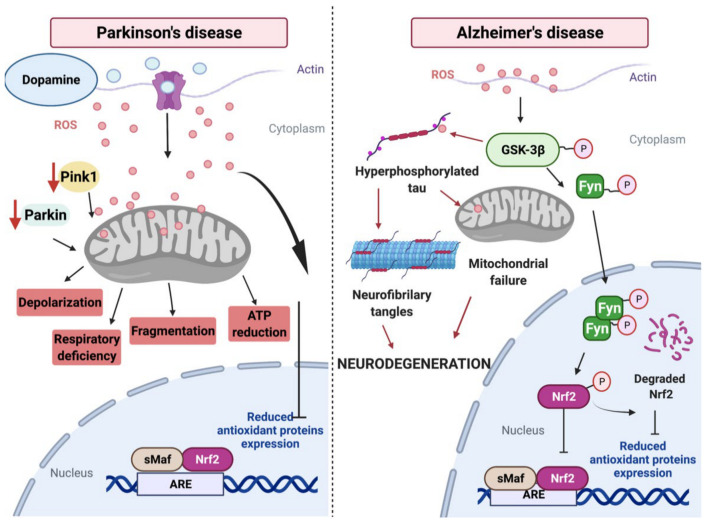
Contribution of the Nrf2 pathway to the pathogenesis of Parkinson’s disease and Alzheimer’s Disease. In PD, an increase in the dopamine release could affect mitochondrial function, producing an increase in ROS levels affecting Nrf2 activity and the response against the oxidative damage. Additionally, the decrease in Parkin and PINK expression levels shown in PD could affect mitochondrial function, inducing depolarization, fragmentation, respiratory deficiency, and ATP reduction. These changes will affect synaptic function, contributing to neurodegeneration and cognitive impairment present in PD. In AD, the GSK-3β protein, which is a kinase that promotes the anomalous phosphorylation of tau protein, promotes Nrf2 degradation by proteasome activity through the Fyn’s phosphorylation. Additionally, during AD, activated GSK-3β induces tau hyperphosphorylation, which could affect mitochondrial function. Later on, the accumulation of pathological forms of tau could lead to the formation of neurofibrillary tangles (NFTs), which is considered a hallmark in AD.

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
