# Peer review of "Contribution of the Nrf2 Pathway on Oxidative Damage and Mitochondrial Failure in Parkinson and Alzheimer’s Disease"

_antioxidants, 2021, doi:10.3390/antiox10071069_

Round 1
Reviewer 1 Report
This review aims to discuss the mechanisms involved in Nrf2 activity and its connection with mitochondria, energy supply, and antioxidant response in PD and AD. The article is well written and gives an interesting historical and scientific perspective. However, the role of Nrf2/ARE Pathway in neurodegeneration has been well known according to previous literature. (PMID: 31964153, PMID: 33144124, PMID: 21403858, PMID: 32714148, etc.) ROS and mitochondria dysfunctions have also been linked to neurodegenerative disorders. (PMID: 24252804, PMID: 31829802, Antioxidants 2021, 10, 794. https://doi.org/10.3390/antiox10050794, PMID: 30675901, PMID: 33679375, etc.) I have some suggestions and recommendations as follows:
- “Here, p62 interacts with the Nrf2-binding site on Keap1, …….increasing free p62 levels “ line 229-230, please check this description again.
- The role and pathogenesis of ROS in PD and AD should be a clear explanation in a subsection.
- “Various vegetable components like l SFN from broccoli…..”, please check again.
- In the part of therapy, the authors only introduced SFN which is an activator of Nrf2. However, there are several kinds of this like compounds under exploration in past researches, for example, CDDO-Me, RTA-408, Dimethyl fumarate, ALKS-8700, Ursodiol, Sulforadex, ITH12674, Curcumin, Resveratrol, NRF2-KEAP1 protein-protein interaction inhibitors such as tetrahydroisoquinoline, and KEAP1-independent activators of NRF2 such as GSK-3 inhibitors. More candidates can be introduced in the text.
Author Response
Reviewer 1
This review aims to discuss the mechanisms involved in Nrf2 activity and its connection with mitochondria, energy supply, and antioxidant response in PD and AD. The article is well written and gives an interesting historical and scientific perspective. However, the role of Nrf2/ARE Pathway in neurodegeneration has been well known according to previous literature. (PMID: 31964153, PMID: 33144124, PMID: 21403858, PMID: 32714148, etc.) ROS and mitochondria dysfunctions have also been linked to neurodegenerative disorders. (PMID: 24252804, PMID: 31829802, Antioxidants 2021, 10, 794. https://doi.org/10.3390/antiox10050794, PMID: 30675901, PMID: 33679375, etc.) I have some suggestions and recommendations as follows:
-Answer. We thank the reviewer for his positive comments. We agree that several authors have suggested an essential role of Nrf2, ROS, and mitochondrial dysfunction in the pathogenesis of neurodegenerative diseases. This is a critical topic because these changes are produced in the early stages of these diseases and can be used as a valid strategy to ameliorate the neurodegenerative changes present in these disorders. However, our manuscript focuses on the changes shown in the two most recognized neurodegenerative diseases (PD and AD) worldwide. As far as we know, very few papers are centered on analyzing the Nrf2 pathway in PD and AD. Also, novel aspects of Nrf2 on AD pathogenesis are discussed; specifically, we examine evidence where Nrf2 reduces the accumulation of tau pathology by activating proteasome activity and reduced mitochondrial dysfunction in AD. These are important observations because pathological forms of tau are centered in AD pathology, and mitochondrial injury is considered a hallmark in the pathogenesis of AD. Also, we included a detailed discussion of the different strategies to activate this pathway which have positive consequences reducing neurodegeneration in these diseases. Therefore, we believe that our work covers aspects of the Nrf2 pathway in SNC that other authors have not addressed.
- “Here, p62 interacts with the Nrf2-binding site on Keap1, …….increasing free p62 levels “ line 229-230, please check this description again.
-Answer. We apologize for this issue. This mistake was eliminated, and the manuscript was corrected accordingly.
- The role and pathogenesis of ROS in PD and AD should be a clear explanation in a subsection.
-Answer. We apologize for this issue. We took reviewer suggestions, and we included a separate section describing the role of ROS in PD and AD.
- “Various vegetable components like l SFN from broccoli…..”, please check again.
-Answer. We apologize for this issue. This mistake was corrected accordingly.
- In the part of therapy, the authors only introduced SFN, which is an activator of Nrf2. However, there are several kinds of this like compounds under exploration in past researches, for example, CDDO-Me, RTA-408, Dimethyl fumarate, ALKS-8700, Ursodiol, Sulforadex, ITH12674, Curcumin, Resveratrol, NRF2-KEAP1 protein-protein interaction inhibitors such as tetrahydroisoquinoline, and KEAP1-independent activators of NRF2 such as GSK-3 inhibitors. More candidates can be introduced in the text.
-Answer. We took reviewer suggestions, and as an example, we included additional evidence describing the action of curcumin, DMF, against oxidative damage and mitochondrial injury in AD. Also, the role of GSK-3b on Nrf2 activation was discussed, and the manuscript was updated accordingly.
Reviewer 2 Report
This is an interesting review on Nrf2, mitochondria homeostasis and neurodegenerative conditions such as Parkinson and Alzheimer diseases. A non negligible amount of literature search has been done.
I think this review needs more maturation. The authors should put more in the front their literature research done concerning the mechanisms revolving around Nrf2/mitochondria and their potential involvement in AD and PD. To highlight better their thoughts, they should work a little bit more on their figures: generate figures that reflect the text, and importantly the details in the text.
In addition, The English/grammar style used in the text need to be checked in depth.
Major concerned:
- Line 79-88: This paragraph describes how Nrf2 is regulated, and it refers to Fig1. But all the actors described in this paragraph are not shown in Fig1. The Figure 1 seems to show mainly how Nrf2 activate ARE genes, and actually the regulation of Nrf2. I suggest the authors to either complete this figure1 or to add one that will reflect the text.
- Similarly, line 89-93, paragraph describing the regulation of Nrf2 via Fyn, and refers to Fig1 but all the actors mentioned are missing in Fig1.
- Proposition: A global figure summarizing the role of Nrf2 in the CNS could be helpful to support the ideas in the chapter 2. Then a figure specifically showing Nrf2, mitochondria homeostasis etc and all actors described in the text should be generated in the context of PD, and another one in the context of AD should also be generated.
- Line 206-212: I’m surprised to see in the paragraph a ref#92 on C2C12 cells, which murine skeletal muscle cell lines. I know these cells have been extremely useful to understand the role of mitochondria in muscle, muscle differentiation etc, but I’m not sure this ref is relevant to this paragraph “2.5- Nrf2 and mitochondrial function in the brain”. Is the regulation of Nrf2 activity via a thiol homeostasis disruption in muscle cells relevant for neuronal cells? Or glial cells?
grammar and typos: List not exhaustive, need to go carefully through the text to edit the English and style.
Line 42: space missing at the beginning of the sentence “Nrf2 activation increases the ….”
Line 43: remove the full stop before the ref 9
Line 44: “Consequently, these actions improves” should be “Consequently, these actions improve”
Line 54-56: “Furthermore, studies from Branca and colleagues showed a significant reduction in HO-1 levels in a transgenic mice for AD (APP/PS1).” What are the Ref for these studies, unless it is only ref #16 the authors are refereeing to, in this context, grammar should be corrected. I would also suggest the authors to combine the 2 sentences (line 54-56) in one to increase the dynamic of the text.
Line 57-59: In the sentence “Also, Nrf2 expression is significantly impaired in nigral dopaminergic neurons of PD patients [14], [17], in addition, the reduction of dopaminergic neurons and inflammatory-mediated microglia activation were enhanced in Nrf2 (-/-) knock out mice [18].” Remove “Also” at the beginning of the sentence and use the word “and “ instead of “in addition”.
Line 61: remove “also”
Line 62: space missing before “indeed”
Line 89: “the glycogen synthase kinase-3b 89 (GSK-3b)” acronym already define in the paragraph above, no need to define it again here.
Line 94: space missing before “Fyn”
Line 120: remove “Also”
Line 132: remove the extra space before “study”
Line 133: Add a space before “Nrf2”
Line 153: remove the extra space before “thioredoxins (TRXs)”
Line 208: grammar need to be corrected: “they targeting mitochondria…” should be “they targeted mitochondria”.
Line 210: “as” missing before “shown”
Line 216: Not sure about the significance of “on the contrary” in this sentence: in contrary to C2C12, else?
Line 225: space missing before “In addition”
Author Response
Reviewer 2
This is an exciting review on Nrf2, mitochondria homeostasis and neurodegenerative conditions such as Parkinson and Alzheimer diseases. A non-negligible amount of literature search has been done.
I think this review needs more maturation. The authors should put more in the front their literature research on the mechanisms revolving around Nrf2/mitochondria and their potential involvement in AD and PD. To highlight better their thoughts, they should work a little bit more on their figures: generate figures that reflect the text, and importantly the details in the text.
-Answer. We thank the reviewer for his positive comments. As the reviewer suggests, we corrected the figures and updated our manuscript accordingly.
In addition, The English/grammar style used in the text needs to be checked in depth.
-Answer. We apologize for this issue. All mistakes were corrected, and the manuscript was updated accordingly
Major concerned:
-Line 79-88: This paragraph describes how Nrf2 is regulated, and it refers to Fig1. But all the actors described in this paragraph are not shown in Fig1. The Figure 1 seems to show mainly how Nrf2 activate ARE genes, and actually the regulation of Nrf2. I suggest the authors to either complete this figure1 or to add one that will reflect the text.
-Answer. We thank the reviewer for his positive comments. As the reviewer suggests, we corrected figures 1 and 2 and updated our manuscript text accordingly.
-Similarly, line 89-93, paragraph describing the regulation of Nrf2 via Fyn, and refers to Fig1 but all the actors mentioned are missing in Fig1.
-Answer. We apologize for this issue. As the reviewer suggests, we corrected figures and updated our manuscript accordingly.
-Proposition: A global figure summarizing the role of Nrf2 in the CNS could be helpful to support the ideas in the chapter 2. Then a figure specifically showing Nrf2, mitochondria homeostasis etc and all actors described in the text should be generated in the context of PD, and another one in the context of AD should also be generated.
-Answer. We apologize for this issue. We corrected Figures 1 and 2 and updated our manuscript text accordingly.
-Line 206-212: I’m surprised to see in the paragraph a ref#92 on C2C12 cells, which murine skeletal muscle cell lines. I know these cells have been extremely useful to understand the role of mitochondria in muscle, muscle differentiation etc, but I’m not sure this ref is relevant to this paragraph “2.5- Nrf2 and mitochondrial function in the brain”. Is the regulation of Nrf2 activity via a thiol homeostasis disruption in muscle cells relevant for neuronal cells? Or glial cells?
-Answer. We apologize for this issue. To avoid clarity issues, we removed evidence related to C2C12 cells.
-grammar and typos: List not exhaustive, need to go carefully through the text to edit the English and style.
-Answer. We apologize for this issue. These mistakes were corrected accordingly
-Line 42: space missing at the beginning of the sentence “Nrf2 activation increases the ….”
Answer. We apologize for this issue. This mistake was corrected accordingly.
-Line 43: remove the full stop before the ref 9
Answer. We apologize for this issue. This mistake was corrected accordingly
-Line 44: “Consequently, these actions improves” should be “Consequently, these actions improve”
-Answer. We apologize for this issue. This mistake was corrected accordingly
Line 54-56: “Furthermore, studies from Branca and colleagues showed a significant reduction in HO-1 levels in a transgenic mice for AD (APP/PS1)." What are the Ref for these studies, unless it is only ref #16 the authors are referring to, in this context, grammar should be corrected. I would also suggest the authors combine the 2 sentences (line 54-56) in one to increase the dynamic of the text.
-Answer. We apologize for this issue. We took reviewer suggestions, and we updated references, corrected grammar, and combined these two sentences.
-Line 57-59: In the sentence “Also, Nrf2 expression is significantly impaired in nigral dopaminergic neurons of PD patients [14], [17], in addition, the reduction of dopaminergic neurons and inflammatory-mediated microglia activation were enhanced in Nrf2 (-/-) knock out mice [18].” Remove “Also” at the beginning of the sentence and use the word “and “ instead of “in addition”.
-Answer. We apologize for this issue. This mistake was corrected accordingly
-Line 61: remove “also”
-Answer. We apologize for this issue. This mistake was corrected accordingly
-Line 62: space missing before “indeed”
-Answer. We apologize for this issue. This mistake was corrected accordingly
-Line 89: “the glycogen synthase kinase-3b 89 (GSK-3b)” acronym already define in the paragraph above, no need to define it again here.
-Answer. We apologize for this issue. This mistake was corrected accordingly
-Line 94: space missing before “Fyn”
-Answer. We apologize for this issue. This mistake was corrected accordingly
-Line 120: remove “Also”
-Answer. We apologize for this issue. This mistake was corrected accordingly
-Line 132: remove the extra space before “study”
-Answer. We apologize for this issue. This mistake was corrected accordingly
-Line 133: Add a space before “Nrf2”
-Answer. We apologize for this issue. This mistake was corrected accordingly
-Line 153: remove the extra space before “thioredoxins (TRXs)”
-Answer. We apologize for this issue. This mistake was corrected accordingly
-Line 208: grammar need to be corrected: “they targeting mitochondria…” should be “they targeted mitochondria”.
-Answer. We apologize for this issue. This mistake was corrected accordingly
-Line 210: “as” missing before “shown”
-Answer. We apologize for this issue. This mistake was corrected accordingly
-Line 216: Not sure about the significance of “on the contrary” in this sentence: in contrary to C2C12, else?
-Answer. We apologize for this issue. This mistake was corrected, and this evidence was removed from the manuscript.
-Line 225: space missing before “In addition”
-Answer. We apologize for this issue. This mistake was corrected accordingly
Reviewer 3 Report
Review of the manuscript entitled: “Contribution of the Nrf2 pathway on oxidative damage and mitochondrial failure in Parkinson and Alzheimer's disease”. The Authors undertaken important topic concerning Nrf2 pathway in the mitochondrial impairment and neurodegeneration present in PD and AD. Overall the work is well prepared but there are some remarks that could improve the paper.
In line 55. Abbreviation “APP/PS1” should be explained.
In line 56. “Aβ deposition and increasing of” the font should be corrected.
According to international guidelines, genes should be in italics.
Line 117-118 AhR is also involved in other processes. Especially in the brain, it is involved in interactions with the PPAR gamma and interactions with extracellular matrix. See papers 10.1007/s00210-018-1591-4 and 10.1016/j.cyto.2019.154930 it may be useful, especially in the context of neurodegenerative diseases.
Line 176 - for me, there is no information about PPARgamma at this paper, which in the metabolism of astrocytes together with AhR and Nrf2 is crucial.
Chapters 1 and 2 the manuscript is about neurodegenerative diseases; therefore, Authors should clearly justify why they write about astrocytes. I know why but it is not emphasized enough, ie. the role of astrocytes in these diseases.
Line 361-362 PPARgamma appears, but should be connected with Nrf2, AhR, ROS and maybe in my opinion crucial ECM proteins
Line 404: correct the font
Author Response
Reviewer 3
Review of the manuscript entitled: "Contribution of the Nrf2 pathway on oxidative damage and mitochondrial failure in Parkinson and Alzheimer's disease". The Authors undertaken important topic concerning the Nrf2 pathway in the mitochondrial impairment and neurodegeneration present in PD and AD. Overall the work is well prepared, but there are some remarks that could improve the paper.
-In line 55. Abbreviation “APP/PS1” should be explained.
-Answer. We apologize for this issue. This mistake was corrected, and information about APP/PS1 mice strain was added.
-In line 56. “Aβ deposition and increasing of” the font should be corrected.
-Answer. We apologize for this issue. This mistake was corrected accordingly
-According to international guidelines, genes should be in italics.
-Answer. Answer. We apologize for this issue. This mistake was corrected accordingly
-Line 117-118 AhR is also involved in other processes. Especially in the brain, it is involved in interactions with the PPAR gamma and interactions with the extracellular matrix. See papers 10.1007/s00210-018-1591-4 and 10.1016/j.cyto.2019.154930 it may be useful, especially in the context of neurodegenerative diseases.
-Answer. We thank the reviewer for his suggestions. In this case, we focus on AhR and Nrf2, and we excluded additional evidence that links AhR with other pathways involved in mitochondrial biogenesis like PPARg.
-Line 176 - for me, there is no information about PPARgamma at this paper, which in the metabolism of astrocytes together with AhR and Nrf2 is crucial.
-Answer. We agree with the reviewer about the critical part that astrocytes and glial cells play in the pathogenesis of PD and AD. However, in the current format is impossible to fit an entire dedicated chapter discussing the role of these cells in the impairment of the Nrf2 pathway in these diseases. We choose to empathize with a few studies describing glial cell's actions promoting neuroinflammation and the NF-Kb pathway impairment present in PD.
-Chapters 1 and 2 the manuscript is about neurodegenerative diseases; therefore, Authors should clearly justify why they write about astrocytes. I know why but it is not emphasized enough, ie. the role of astrocytes in these diseases.
-Answer. We agree with the reviewer about the critical part that astrocytes and glial cells play in the pathogenesis of PD and AD. However, it is difficult to present evidence from neurons and glial cells separately in our manuscript's present format.
-Line 361-362 PPARgamma appears, but should be connected with Nrf2, AhR, ROS, and maybe in my opinion crucial ECM proteins
-Answer. We agree with the reviewer about the connection between Nrf2 and PPARgsignaling. Evidence indicates that both pathways promote mitochondrial biogenesis in a complementary way in the brain. We are aware of the importance of the PPARgpathway in the pathogenesis of several neurodegenerative diseases, including AD, PD, and HD. Our group previously published two interesting reviews (Therapeutic Actions of the Thiazolidinediones in Alzheimer's Disease, PPAR Res 2015; 2015:957248.DOI: 10.1155/2015/957248; Role of PPAR γ in the Differentiation and Function of Neurons, PPAR Res 2014;2014:768594.doi: 10.1155/2014/768594) that explored the contribution of this pathway on mitochondrial dysfunction present in neurodegenerative diseases (HD, PD, AD) and its potential role of PPARgon neuronal function and differentiation. However, it is not very easy to include this evidence in the present manuscript format because it would increase manuscript length beyond journal standards.
-Line 404: correct the font
-Answer. We apologize for this issue. This mistake was corrected accordingly
Round 2
Reviewer 2 Report
The authors answered to all my concerns, and improved the paper. Just need to check few edition problems (mostly space missing at beginning of sentences). This can be done at the edition stage.
Author Response
The authors answered to all my concerns, and improved the paper. Just need to check few edition problems (mostly space missing at beginning of sentences). This can be done at the edition stage.
Answer. Thanks to the reviewer for their comments. We updated our manuscript and corrected all format mistakes.